# Potential of Genome-Wide Association Studies and Genomic Selection to Improve Productivity and Quality of Commercial Timber Species in Tropical Rainforest, a Case Study of *Shorea platyclados*

**Sawitri [1,2], Naoki Tani [3,4,]***, **Mohammad Na'iem [2], Widiyatno [2], Sapto Indrioko [2], Kentaro Uchiyama [5], Rempei Suwa [3], Kevin Kit Siong Ng [6], Soon Leong Lee [6] and Yoshihiko Tsumura [4,]***

[1]  Graduate School of Life and Environmental Sciences, University of Tsukuba, Tennodai 1-1-1, Tsukuba, Ibaraki 305-8572, Japan; sawitriforester@gmail.com
[2]  Faculty of Forestry, Gadjah Mada University, Bulaksumur, Yogyakarta 55281, Indonesia; moh_naiem@yahoo.com (M.N.); widiyatno.yk@gmail.com (W.); saptoindrioko@yahoo.com (S.I.)
[3]  Forestry Division, Japan International Research Center for Agricultural Sciences, Ohwasi 1-1, Tsukuba, Ibaraki 305-8686, Japan; swrmp@affrc.go.jp
[4]  Faculty of Life and Environmental Sciences, University of Tsukuba, Tennodai 1-1-1, Tsukuba, Ibaraki 305-8572, Japan
[5]  Department of Forest Molecular Genetics and Biotechnology, Forestry and Forest Products Research Institute, 1 Matsunato, Tsukuba, Ibaraki 305-8687, Japan; kruchiyama@affrc.go.jp
[6]  Forestry Biotechnology Division, Forest Research Institute Malaysia, Kepong 52109, Selangor Darul Ehsan, Malaysia; kevin@frim.gov.my (K.K.S.N.); leesl@frim.gov.my (S.L.L.)
**\***  Correspondence: ntani@affrc.go.jp (N.T.); tsumura.yoshihiko.ke@u.tsukuba.ac.jp (Y.T.); Tel.: +81-298386316 (N.T.); +81-298534629 (Y.T.)

**Abstract:** *Shorea platyclados* (Dark Red Meranti) is a commercially important timber tree species in Southeast Asia. However, its stocks have dramatically declined due, *inter alia*, to excessive logging, insufficient natural regeneration and a slow recovery rate. Thus, there is a need to promote enrichment planting and develop effective technique to support its rehabilitation and improve timber production through implementation of Genome-Wide Association Studies (GWAS) and Genomic Selection (GS). To assist such efforts, plant materials were collected from a half-sib progeny population in Sari Bumi Kusuma forest concession, Kalimantan, Indonesia. Using 5900 markers in sequences obtained from 356 individuals, we detected high linkage disequilibrium (LD) extending up to >145 kb, suggesting that associations between phenotypic traits and markers in LD can be more easily and feasibly detected with GWAS than with analysis of quantitative trait loci (QTLs). However, the detection power of GWAS seems low, since few single nucleotide polymorphisms linked to any focal traits were detected with a stringent false discovery rate, indicating that the species' phenotypic traits are mostly under polygenic quantitative control. Furthermore, Machine Learning provided higher prediction accuracies than Bayesian methods. We also found that stem diameter, branch diameter ratio and wood density were more predictable than height, clear bole, branch angle and wood stiffness traits. Our study suggests that GS has potential for improving the productivity and quality of *S. platyclados*, and our genomic heritability estimates may improve the selection of traits to target in future breeding of this species.

**Keywords:** *Shorea platyclados*; GWAS; GS; growth; wood quality traits; genomic heritability

## 1. Introduction

Tropical rainforests in Southeast Asia are dominated by Dipterocarpaceae family [1,2], including *Shorea platyclados*, which has excellent timber quality [3]. In international trade, it is one of the commercially important timber tree species known as Dark Red Meranti. However, increasing global demand for its timber products triggered proliferation of excessive logging activities in the natural forests [4]. This, together with insufficient natural regeneration and a slow recovery rate has led to a dramatic decline in its stocks [5,6]. Thus, there are needs to promote enrichment planting and develop effective techniques to support forest rehabilitation and improve good quality timber production. For these reasons, *S. platyclados* breeding programs started in Indonesia in 2006 with collection of numerous seeds from 81 mother trees in natural populations and were then planted in a progeny trial. Phenotypic traits were then evaluated to select the trees that showed the best growth performance. However, due to the long lifespan and juvenility period of forest tree species, evaluation of such conventional breeding is time-consuming, laborious and expensive. Thus, genome-wide approaches, including Genome-Wide Association Study (GWAS) and Genomic Selection (GS), have been recently developed to overcome these problems [7–9]. These techniques seem to offer promising strategies for genetic improvement of complex traits and accelerating breeding cycles of forest tree species [7,10,11]. GWAS provides potential capacity to identify the genes that are causally associated with phenotypes of focal traits, especially traits that are strongly affected by one or a few genes. However, it has been less successful in identifying such genes in species with phenotypes that are mostly influenced by numerous minor quantitative trait loci (QTLs) [12,13].

GS is fundamentally different from GWAS, as it involves use a full-genome information, regardless of its significance, in relation to a specific trait, rather than a few markers as in GWAS. This genotypic information, collected from training and validation population, is used in conjunction with corresponding phenotypic data, collected from training population, to develop a predictive model [12,14]. In forest tree breeding programs, GWAS and GS could substantially reduce the length of breeding cycles and increase genetic gain per unit time through early selection of superior genotypes during the juvenile phase. Therefore, the entire cycles of progeny field testing can potentially be skipped in the selection process [8,15].

Numerous statistical models and algorithms have been developed for improving genomic prediction accuracy, including parametric and non-parametric statistical methods [16,17]. Parametric methods, including Bayesian techniques, such as Bayesian Ridge Regression, Bayesian LASSO, Bayes A, Bayes B and Bayes C [18–20], are the most commonly used in GS studies of forest tree species [21–23]. Since these statistical methods cannot explicitly account for interactions among single nucleotide polymorphisms (SNPs), application of Machine Learning in GS studies has been proposed. Machine Learning is being increasingly applied in GS studies because it does not require any assumptions about the underlying traits, it is easy to use, and it can both capture complex non-linear relationships and efficiently increase prediction accuracy [14]. Popular Machine Learning methods include Random Forest (RF), Extreme Gradient Boosting (XgBoost) and Bayesian Additive Regression Tree (BART) modelling. In RF modelling, many decision trees (often hundreds to thousands) are constructed and de-correlated so the average over the resulting 'forest ensemble' will result in lower variance. XgBoost involves similar principles, but applies a more regularized model to control over-fitting. BART is a sum-of-trees method, in which each decision tree is constrained by three Bayesian regularization prior distribution. A study of beef cattle has confirmed that Machine Learning methods can successfully improve the prediction accuracy of focal traits [24–26]. Beside statistical model performance, other factors are known to influence the accuracy of genomic prediction, such as the extent and distribution of linkage disequilibrium between markers and QTLs, trait heritability, marker density and relationship between training and validation population [15,27].

GWAS and GS have been successfully applied in numerous evaluations of genetic controls of growth, wood properties, disease resistance and male fecundity in forest tree species including *Eucalyptus grandis* × *E. camaldulensis* [15,23], *E. pellita*, *E. benthamii* [22], *E. grandis* [28], *E. grandis* × *E.*

*urophylla* [29], loblolly pine (*Pinus taeda*) [23], white spruce (*Picea glauca*) [30], maritime pine (*Pinus pinaster*) [31] and recently Japanese cedar (*Cryptomeria japonica*) [32,33]. Therefore, the study presented here had four objectives. First, to assess the extent of linkage disequilibrium (LD) in a half-sib progeny trial population of *S. platyclados*. Second, to apply GWAS to assess its utility for enhancing the timber yield (tree growth) and timber quality (wood density and stiffness). Third, to evaluate the efficiency of genomic prediction using Bayesian methods and three Machine Learning methods. Fourth, to estimate the genomic heritability of the considered traits that may improve the genomic prediction accuracy.

## 2. Materials and Methods

### 2.1. Study Site

The study focused on material in a progeny trial of a *S. platyclados* population at Sari Bumi Kusuma forest concession (PT-SBK), Central Kalimantan, Indonesia (Figure 1). The material consists of representatives of 81 open-pollinated half-sib families collected (as seeds) from mother trees in a natural population in the PT-SBK region. The trial was established in May 2006 using a factorial Randomized Complete Block Design (RCBD) with nine replications, each containing four individuals from a single mother tree. Seedlings were planted with 6 × 3 m initial spacing between them. PT-SBK management has included two thinnings to increase spacing, remove unhealthy and damaged individuals, and reduce competitional effects of neighboring trees. During the progeny trial establishment, individuals from four mother trees did not survive, thus 77 mother trees remained. In September 2017, a total of 420 *S. platyclados* trees were selected from four replicates of the progeny trials that had a mean, minimum and maximum number of offspring per family of 5, 1 and 10 respectively. They were then phenotyped and genotyped for the analysis presented here. After sequencing and filtering, data pertaining to 356 individuals were subjected to GWAS and GS analysis.

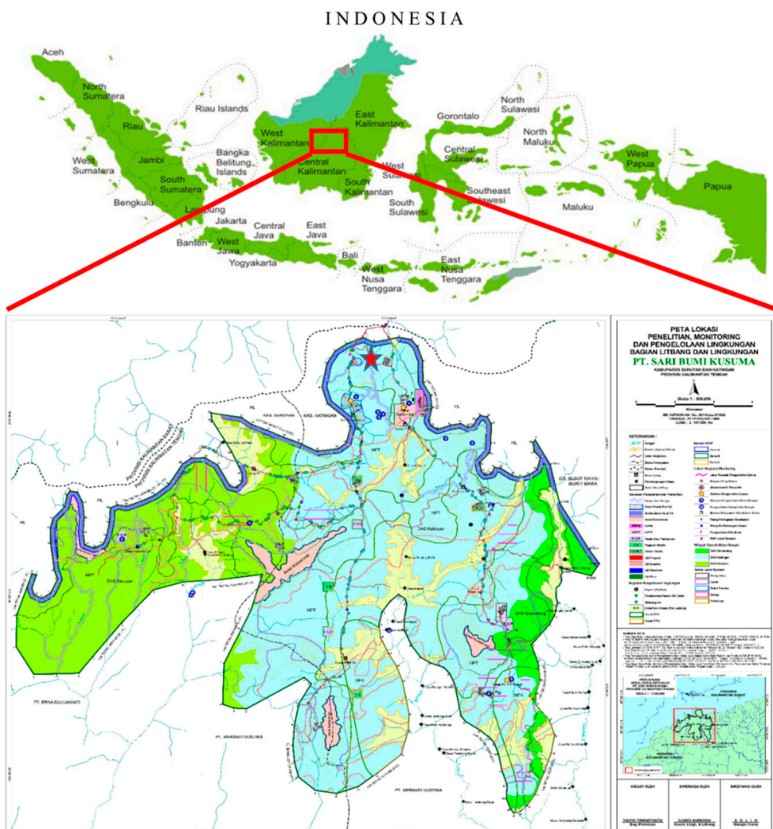

**Figure 1.** *Shorea platyclados* half-sib progeny trial research site at SBK forest concession, Central Kalimantan, Indonesia.

## 2.2. Phenotypic Data

In 2017, 11 years after planting (after the second thinning), we measured seven important traits, including growth, branching quality and wood quality traits of the 420 *S. platyclados* trees. The growth traits were stem diameter at breast height (DBH), total height and clear bole height; the branching quality traits were branch angle and branch diameter ratio, and the wood quality properties were wood density and wood stiffness. We also measured DBH and total height in 2014, eight years after planting (before thinning). Stem diameter at breast height was measured at 1.3 m above ground level using a diameter tape. Total height was measured from the ground to the top of the tree using Haga altimeter, while clear bole was defined as the height of the tree from stem base to the crown base. Branch angle was defined as the vertical intersection angle between the pith of the first branch of the tree's crown and the stem, recorded in four categorial scores: 1 ($\leq$22.5°), 2 (>22.5–45°), 3 (>45–67.5°) and 4 (>67.5–90°). The branch diameter ratio (the ratio of the diameter of this branch to the stem diameter at the branching point) was recorded in six categorical scores: 1 (1:1), 2 (2:1), 3 (3:1), 4 (4:1), (5:1) and 6 (6:1). Wood density and wood stiffness were measured using Pilodyn 6J Forest (FTS., Ltd., Tokyo, Japan) and TreeSonic Microsend Timer (Fakopp Enterprise Bt., Fenyo, Hungary) instruments, respectively.

## 2.3. Genotypic Data and SNP Discovery

Leaf samples were taken from each selected individual, then stored at −20 °C prior to DNA extraction. Total genomic DNA was extracted from 20 mg of each sample using a modified cetyltrimethyl ammonium bromide (CTAB) method [34]. After purifying the extracted DNA using a Chromatin Immunoprecipitation (ChIP) DNA Clean & Concentrator$^{TM}$-10 Kit (Zymo Research, Irvine, CA, USA), its concentration was quantified using a Qubit$^{TM}$ dsDNA Broad Range Assay Kit (Thermo Fisher Scientific, Waltham, MA, USA). Double digest restriction site-associated DNA sequencing (ddRADseq) libraries were prepared following [35]. Briefly, genomic DNAs were double digested using *Pst*I and *Sau3A*I restriction enzymes (Invitrogen, Waltham, MA, USA), ligated with Y-shaped adaptors and amplified using PCR with KAPA HiFi polymerase (KAPA BIOSYSTEMS, Boston, MA, USA). After PCR amplification with adapter specific primer pairs (Access Array Barcode Library for Illumina, Fluidigm, South San Francisco, CA, USA), an equal amount of DNA from each sample was mixed and size-selected with BluePippin 2% agarose gel (Sage Science, Beverly, MA, USA). Approximately 450 bp library fragments were retrieved. The quality of the library was checked using KAPA Library Quantification Kits on LightCycler 480 Instrument (Roche, Basel, Switzerland). Finally, nucleotide sequence libraries were then sequenced using a high-throughput Illumina Hi-Seq X Ten platform (Macrogen, Inc., Seoul, South Korea) to generate paired-end reads with a 150 bp long.

Out of 420 individuals genotyped in this research, the Illumina Hi-Seq X Ten platform successfully sequenced 384 individuals. Obtained raw reads were mapped to a reference genome sequences data (unpublished data) of *S. leprosula*, which is closely related species of *S. platyclados* [36]. The draft genome sequence was obtained by Ng et al. entitled "Dipterocarp genomes highlight the ecological relevance of drought in a seasonal tropical rainforest" (unpublished observation). In addition, [37] estimated that C-value of *S. platyclados* was 0.412 pg. We obtained a rough estimation of genome size for *S. platyclados* is 379 Mbp. The *dDocent* pipeline [38] was used for quality trimming (Trimmomatic v.0.33) [39], read mapping (BWA mem v.0.7.12) [40] and SNP calling (FreeBayes v.0.9.20) [41]. We selected loci (sites) that were biallelic (without indels in both reference and samples) and polymorphic in the samples, with higher sequencing quality and fewer missing genotypes using the VCFtools [42]. Initially, a total of 643 million reads (643,464,580 reads) covering 97.16 Gb of sequence data, with an average of 1.2 million reads (1,675,689 reads) per sample were generated. Furthermore, we excluded 17 individuals due to low quality reads sequencing (less than 100,000 reads), and thus, 367 individuals remained with a total of 643,446,244 sequencing reads (97.16 Gb). In the first filtering step we only retained variants (257,545 SNPs) that had been successfully genotyped in more than 50% of the individuals (max missing parameter: 0.5), with minimum allele count of 3 (mac 3), minimum quality score of 30 (minQ 30) and minimum sequencing depth per SNP and individual of 3 (minDP3). The next step was

to eliminate data pertaining to individuals that had not been sequenced well by assessing individual levels of missing data with VCFtools [42], thus, sequences of 356 individuals remained. An iterative filtering process was then applied to maximize the number of individuals and loci in the final dataset by applying the following criteria: calling rate > 95%, minor allele frequency (MAF) < 0.05 and minDP < 10. This reduced the total number of SNPs to 27,829. We then filtered loci based on numerous criteria, including allele balance at heterozygous loci, overlapping forward and reverse reads, proper read pairing and ratio of quality depth. It remained 8831 SNPs. We then removed SNPs that deviated significantly from Hardy-Weinberg Equilibrium (HWE) at $p > 0.001$. After these entire filtering process, a total of 5900 high quality SNPs of 356 individuals were retained for the GWAS and GS analyses.

## 2.4. Population Structure and Genetic Diversity

LD-based SNP pruning was implemented in PLINK v1.9 [43] to select only SNPs that are approximately uncorrelated with each other based on these criteria: window size in SNPs was 1000, the number of SNPs to shift the window at each step was five and the variance inflation factor (VIF) threshold was 1.5. This step resulted 2660 SNPs. Population structure and genetic diversity was then calculated among 356 individuals and 2660 SNP markers of *S. platyclados*. Population structure analyses was estimated based on Probabilistic Principal Component Analysis (PPCA) performed using "ppca" function in the pcaMethods package in R [44]. Furthermore, we estimated the genetic diversity using the program GenAlEx v.6.5 [45], including the mean number of alleles (*Na*), number of effective alleles (*Ne*), Shannon's index (*I*), diversity index (*h*), unbiased diversity index (*uh*) and percentage of polymorphic loci (*PPL*).

## 2.5. Linkage Disequilibrium

Linkage Disequilibrium (LD) was estimated by calculating the squared allele frequency correlation coefficient ($r^2$) between pairs of 5900 SNP markers distributed throughout the genome using the TASSEL 5.0 software package [46]. The $r^2$ values were plotted against corresponding genetic distances in base pairs (bp). A smooth line was drawn using second-degree locally weighted polynomial regression (LOESS) by applying the "loess" function in the R statistical program (http://www.r-project.org). The intersection between this line and a threshold value of 0.1 for $r^2$ (plotted as horizontal line in the resulting LD scatterplots) was assumed to provide indications of LD decay in the *S. platyclados* genome [47,48].

## 2.6. Spatial Analysis and Genotype Imputation

As already mentioned, an RCBD experimental design was applied in the progeny trial, in accordance with its frequent use in trials covering large physical areas with numerous individuals, including multiple individuals from each family, and large spacing between individuals, with significant between-microsite variation [49]. This experimental design and environmental variation lead to significant block (fixed) and spatial (random) effects on phenotypic performance. Therefore, spatial analysis was necessary before GWAS and GS to account for these effects. For this, a spatial autocorrelation structure (AR1 × AR1) model, implemented in breedR packages of R [50,51], was applied.

We also imputed missing marker genotypes to replace missing observations in the dataset and boost the statistical power of genomic prediction [52,53], using BEAGLE 4.1 [54]. The algorithm run in this pipeline started with randomly phasing genotypes and imputing missing values of individuals. An iterative "expectation-maximization" algorithm update was repeated in a subsequent step for re-estimating phases and re-inferring missing values from current sampling of phasing information [55]. The imputed genotype data were then subsequently used in GWAS and genomic prediction analyses.

## 2.7. GWAS Using All Individuals and Markers

Data on 5900 markers in 356 individuals were used for GWAS analysis, in which a false discovery rate (FDR) of 0.05 [56] was applied to detect statistically significant results [57] implemented in rrBLUP package in R [58]. To determine the influence of population stratification and kinship relationship, we compared four models and examined the distribution of *p*-values obtained in the association test, including a mixed model with no structure or kinship effects (naïve model), model with population structure (Q model), model with covariates to account for kinship effects (K model) and a mixed model that incorporated both population structure and marker-based kinship estimation (Q+K model). Q values (FDR-corrected *p* values) were calculated using the "p.adjust" function in R. Quantile-quantile (Q-Q) and Manhattan plots were generated with the qqman in R package [59] to assess for the model's ability to control for type I errors and identify significant genes explaining phenotypic variation.

## 2.8. Four-Fold Cross-Validation of GWAS-Based Genomic Prediction

We randomly split 356 individuals into four sets of 89, then used each combination of three sets as training populations, and the other ones as a validation population for four-fold cross validation. We then applied GWAS using 5900 SNPs of all individuals in the training population, which were obtained with the "GWAS" function implemented in the rrBLUP package in R [58]. In the GWAS analysis, $-\log_{10}(P)$ value of each marker for each of the focal traits was calculated. A Mixed Linear Model was applied to reduce spurious associations or false positive (type I errors) by computing a kinship matrix (K) and account for population genetic structure (Q). The kinship matrix was computed by the "A.mat" function of the rrBLUP package in R, while the "n.PC" option was used to specify the number of principal components (PCs) as fixed-effect covariates to account for population structure [33,60]. PCs were estimated by Probabilistic Principal Component Analysis (PPCA). Following previous studies [21,33], with modification, we used two datasets to assess genomic prediction accuracy:

(a) all SNP markers in the whole genome (5900 SNPs);
(b) selected SNP markers with high $-\log_{10}(P)$ values according to the GWAS analysis (applying five GWAS-based thresholds: $-\log_{10}(P) > 0.5$, $-\log_{10}(P) > 0.75$, $-\log_{10}(P) > 1$, $-\log_{10}(P) > 1.25$, and $-\log_{10}(P) > 1.5$.

Genomic prediction accuracy was estimated using both parametric and non-parametric methods. The parametric methods included use of five Bayesian models: Bayesian LASSO, Bayesian Ridge Regression, Bayes A, Bayes B and Bayes C implemented in the BGLR Package in R [20,22]. The non-parametric methods included three Machine Learning methods: RF, XgBoost and BART [24–26]. We tested the accuracy of each model for predicting phenotypic traits for the validation population using four-fold cross validation (with 267 individuals as the training set and 89 remaining individuals as the validation set in each fold). Prediction accuracy was estimated as the correlation between the predicted Genomic Estimated Breeding Values (GEBVs) and observed phenotypes in all the cross-validation tests. We then estimated the genomic prediction accuracy using Bayesian models for all subsets. Following this step, we selected the best $-\log_{10}(P)$ threshold providing the highest prediction accuracy and lowest Deviance Information Criterion (DIC). Finally, we used the selected subset of SNPs to estimate the genomic prediction accuracy provided by the Machine Learning methods (RF, XgBoost and BART).

## 2.9. Genomic Heritability

Narrow-sense genomic heritability, defined as the proportion of additive variance that can be explained by linear regression on a set of markers, was calculated as the ratio of additive genetic variance to the total phenotypic variance using the equation: $h^2 = \sigma_a^2/\sigma_y^2$ [61]. Genomic heritability was obtained from variance components estimated using the Bayesian Ridge Regression model (which resulted in the lowest DIC value for all traits in the genomic prediction analysis) implemented in the BGLR R-package [20]. We calculated genomic heritability using all markers and selected markers. The term "selected markers" was defined as the markers that have $-\log10(P) > 0.75$ for stem diameter

and −log10(P) > 0.5 for height traits at age 8 years, −log$_{10}$(P) > 1, −log$_{10}$(P) > 0.5 for stem diameter and height traits at age 11 years, −log$_{10}$(P) > 0.5, −log$_{10}$(P) > 1.25, −log$_{10}$(P) > 1.25 for clear bole, branch angle and branch diameter ratio, respectively. In addition, markers that have −log$_{10}$(P) > 0.5 and −log$_{10}$(P) > 1 were selected to calculate genomic heritability of wood density and wood stiffness. These −log$_{10}$(P) values were chosen based on their ability in resulting the highest prediction ability represented in Bayesian models.

## 3. Results

### 3.1. Population Structure and Linkage Disequilibrium

Population structure across *S. platyclados* individuals was assessed by PPCA with six axes. The first two PCs explained 3.58% and 2.70% of the total genetic variance (Figure 2). The population formed a single group; mean number of alleles (*Na*) and number of effective alleles (*Ne*) were 1.843 and 1.269, respectively; average Shannon's (*I*), diversity (*h*) and unbiased diversity index (*uh*) had values of 0.324, 0.192 and 0.195, respectively; the percentage of polymorphic loci (*PPL*) in the population was 92.14% (Table S1). Examination of the pattern of physical LD over 5900 SNPs of the 356 genotypes showed that the genome-wide average $r^2$ dropped below 0.1 within 145 kb (Figure 3).

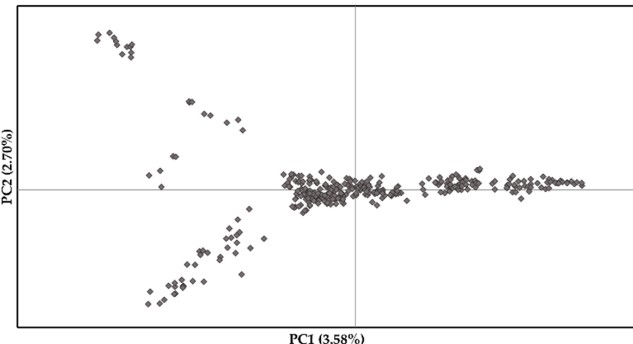

**Figure 2.** First two principal coordinates revealing the population structure of the *S. platyclados* half-sib progeny population.

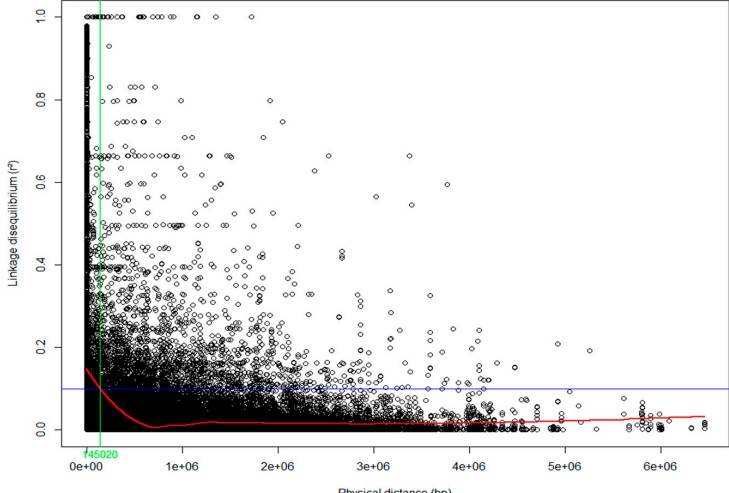

**Figure 3.** Linkage disequilibrium (LD) decay in the *S. platyclados* half-sib progeny population. Pairwise LD ($r^2$) values plotted against the physical distance (bp) between all pairs of single nucleotide polymorphisms (SNPs). The trend line of the nonlinear regressions against physical distance is given by the red line. The horizontal blue and vertical green lines represent the critical value of $r^2$ (0.1) and LD decay value, respectively.

## 3.2. Genome-Wide Association Study

A mixed linear model (MLM), accounting for both population structure (Q) and kinship relatedness (K) was used for detecting marker-trait associations in the GWAS. The MLM (Q+K) model yielded the least departures of observed *P*-value from the expected *P*-value distribution in the validation population, as shown in the Q-Q plots for all traits (Figures 4–6). In addition, the observed *p*-value in the naïve model, Q model and K model also showed the least deviation from the expected *p*-value, which indicated that these models may also be able to control false positives (Figures S1–S4). Of the 5900 SNPs tested with two traits before thinning and seven traits after thinning, we identified one significant SNPs of the height traits in age 8 years (sscaffold00183_353635) that have a *P* FDR value of 0.0045. However, we could not identify whether SNPs showed a significant association (FDR-adjusted *p*-value < 0.05) with other traits. The minimum $-\log_{10}(P)$ values for the traits were $6.59 \times 10^{-5}$ and $9.30 \times 10^{-5}$ for DBH and height before thinning, respectively, $4.14 \times 10^{-6}$ and $2.22 \times 10^{-5}$ for DBH and height after thinning, respectively, $7.99 \times 10^{-5}$ for clear bole, $8.74 \times 10^{-5}$ for branch angle, $8.00 \times 10^{-5}$ for branch diameter ratio, $4.35 \times 10^{-5}$ for wood density and $1.49 \times 10^{-4}$ for wood stiffness. The FDR-adjusted *P*-values were 0.71189 to 0.99985 and 0.00454 to 0.99979 for DBH and height before thinning, respectively. After thinning, these values were higher for DBH (0.12991 to 0.99999), 0.34997 to 0.99995 for height, 0.27502 to 0.99982 for clear bole, 0.99979 to 0.99980 for branch angle, 0.27902 to 0.99982 for branch diameter ratio, 0.55730 to 0.99990 for wood density and 0.47262 to 0.99966 for wood stiffness (Table S2).

Stem diameter

Total height

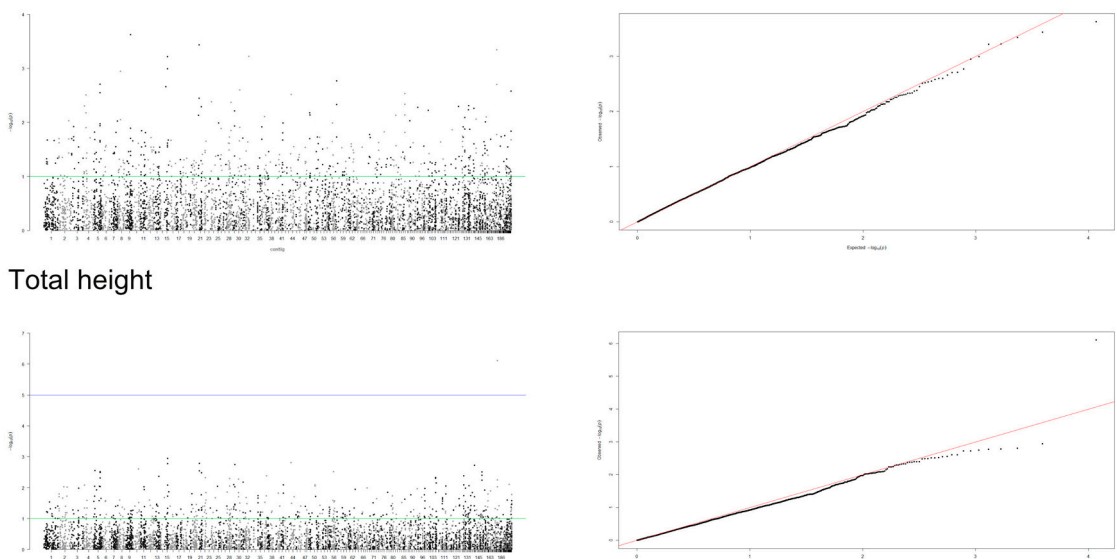

**Figure 4.** Manhattan and Q-Q plots for genome-wide association of *S. platyclados* growth traits (before thinning) with Single Nucleotide Polymorphism (SNP) markers of Q+K model. The quantile-quantile plot indicates the fitness between expected (red lines) and observed *p*-values (black dots).

Stem diameter

Total height

Clear bole height

Branch angle

Branch diameter ratio

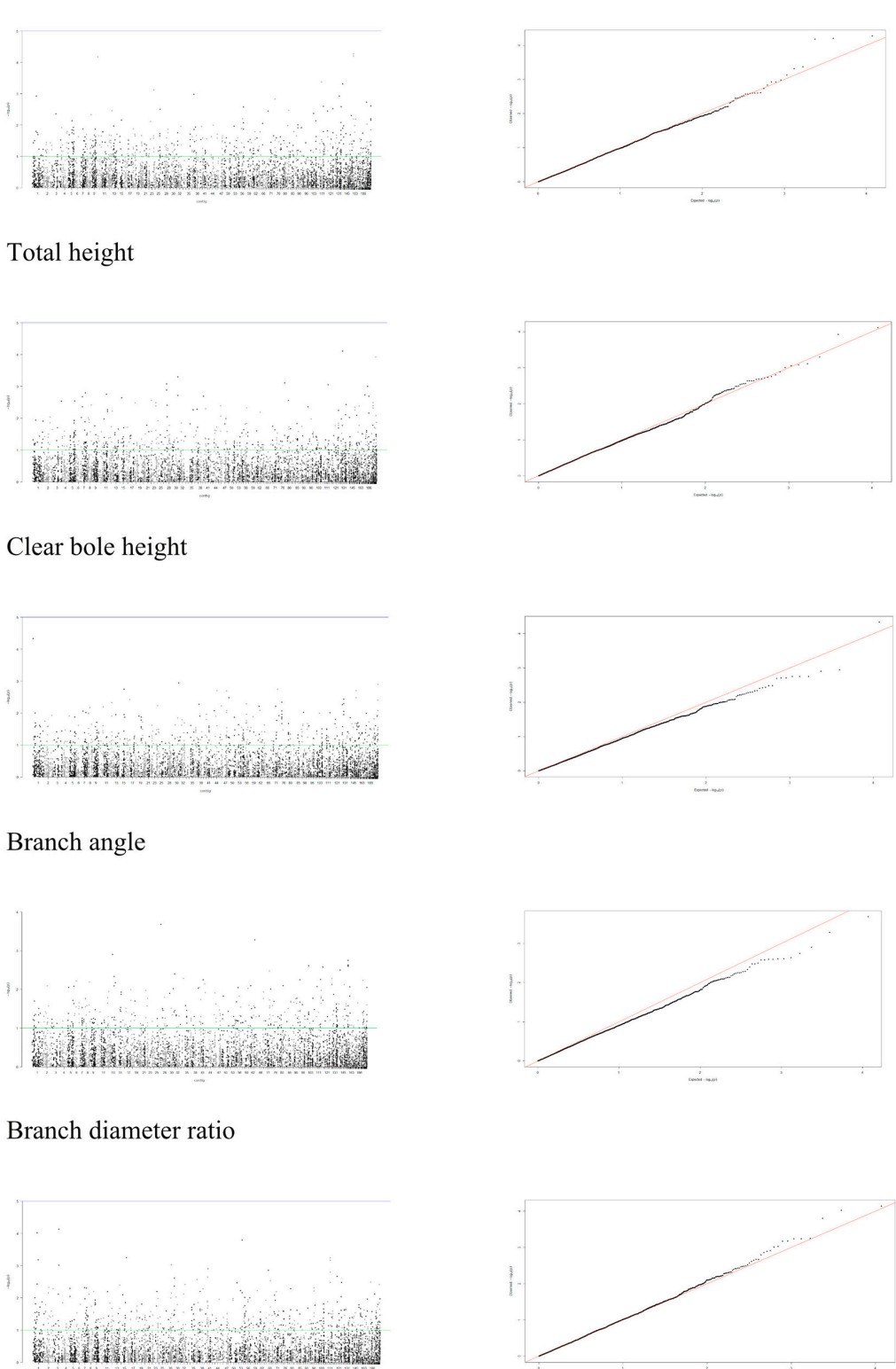

**Figure 5.** Manhattan and Q-Q plots for genome-wide association of *S. platyclados* growth traits (after thinning) and branch quality traits with SNP markers of Q+K model. The quantile-quantile plot indicates the fitness between expected (red lines) and observed *p*-values (black dots).

Wood density

Wood stiffness

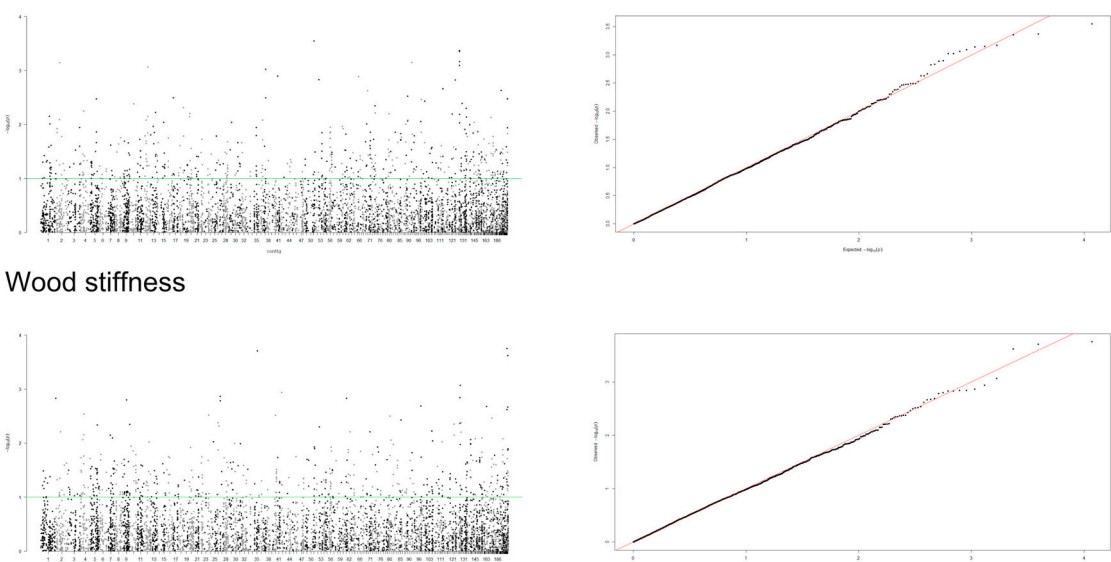

**Figure 6.** Manhattan and Q-Q plots for genome-wide association of *S. platyclados* wood quality traits with SNP markers of Q+K model. The quantile-quantile plot indicates the fitness between expected (red lines) and observed *p*-values (black dots).

In addition, the GWAS identified a number of significant markers based on the of $-\log_{10}(P)$ criteria: $-\log_{10}(P) > 0.5$, $-\log_{10}(P) > 0.75$, $-\log_{10}(P) > 1$, $-\log_{10}(P) > 1.25$ and $-\log_{10}(P) > 1.5$) (Table 1). Moreover, different $-\log_{10}(P)$ criteria resulted in different significant marker densities. When the $-\log_{10}(P) > 0.5$ threshold was applied, the number of SNPs markers dropped from the initial total by approximately 67.69% to 72.39% and further increases in the threshold successively increased the impact on marker density. With the highest threshold, $-\log_{10}(P) > 1.5$, only 196 to 118 SNP markers were retained for each phenotypic trait. Using these total numbers of SNPs, we then tested the effect of each significant marker on the genomic prediction accuracy.

**Table 1.** Numbers of significant SNPs detected and applied in four-fold cross-validation for indicated growth, branch and wood quality traits of *Shorea platyclados* population with indicated Genome-Wide Association Studies (GWAS)-based thresholds.

| Traits | All SNPs | $-\log_{10}(P) > 0.5$ | | | | $-\log_{10}(P) > 0.75$ | | | | $-\log_{10}(P) > 1$ | | | | $-\log_{10}(P) > 1.25$ | | | | $-\log_{10}(P) > 1.5$ | | | |
|---|---|---|---|---|---|---|---|---|---|---|---|---|---|---|---|---|---|---|---|---|---|
| | | Fold1 | Fold2 | Fold3 | Fold4 | Fold1 | Fold2 | Fold3 | Fold4 | Fold1 | Fold2 | Fold3 | Fold4 | Fold1 | Fold2 | Fold3 | Fold4 | Fold1 | Fold2 | Fold3 | Fold4 |
| Stem diameter_2014 | 5900 | 1869 | 1838 | 1906 | 1741 | 1034 | 1011 | 1112 | 975 | 576 | 556 | 566 | 520 | 314 | 331 | 314 | 282 | 163 | 182 | 158 | 135 |
| Total height_2014 | 5900 | 1681 | 1783 | 1711 | 1714 | 926 | 961 | 953 | 906 | 516 | 527 | 515 | 480 | 287 | 301 | 284 | 258 | 155 | 170 | 172 | 140 |
| Stem diameter_2017 | 5900 | 1770 | 1848 | 1807 | 1805 | 995 | 1011 | 980 | 1037 | 534 | 550 | 555 | 539 | 300 | 327 | 295 | 294 | 171 | 177 | 167 | 181 |
| Total height_2017 | 5900 | 1813 | 1810 | 1838 | 1803 | 1023 | 1040 | 1011 | 1010 | 541 | 584 | 583 | 570 | 312 | 312 | 319 | 306 | 181 | 170 | 177 | 169 |
| Clear bole height | 5900 | 1837 | 1760 | 1629 | 1787 | 1043 | 968 | 863 | 995 | 600 | 540 | 487 | 533 | 322 | 290 | 276 | 270 | 172 | 167 | 145 | 157 |
| Branch angle | 5900 | 1709 | 1839 | 1789 | 1665 | 899 | 1008 | 959 | 852 | 488 | 570 | 527 | 444 | 250 | 319 | 265 | 231 | 128 | 192 | 130 | 118 |
| Branch diameter ratio | 5900 | 1847 | 1871 | 1893 | 1841 | 1024 | 1071 | 1022 | 1032 | 594 | 599 | 574 | 590 | 331 | 314 | 327 | 321 | 188 | 187 | 187 | 191 |
| Wood density | 5900 | 1800 | 1807 | 1809 | 1781 | 975 | 987 | 1029 | 992 | 547 | 565 | 606 | 555 | 285 | 320 | 337 | 314 | 171 | 196 | 195 | 171 |
| Wood stiffness | 5900 | 1845 | 1806 | 1785 | 1814 | 1027 | 1007 | 994 | 1003 | 552 | 567 | 554 | 529 | 300 | 331 | 323 | 310 | 151 | 194 | 162 | 186 |

*3.3. Prediction Accuracies Based on the Bayesian Models*

Table 2, Table 3 and Table S3 present results obtained with five commonly used Bayesian models (Bayesian LASSO, Bayesian Ridge Regression, Bayes A, Bayes B and Bayes C) for genomic prediction, including Deviance Information Criterion (DIC) values, which indicate the goodness of fit and model complexity for each trait. All SNP subsets (5900 SNPs) showed low prediction accuracy for height before thinning, clear bole and branch angle traits. Prediction accuracy for stem diameter and height traits before thinning ranged from 0.145 to 0.158 and 0.043 to 0.075, respectively (Table 2), and after thinning prediction accuracies were also low for clear bole height (−0.023 to 0.005) and branch angle (−0.088 to −0.078). However, prediction accuracies for other traits were better: 0.255 to 0.260 for stem diameter, 0.148 to 0.165 for height, 0.220 to 0.230 for branch diameter ratio, 0.220 to 0.233 for wood density and 0.165 to 0.173 for wood stiffness (Table 3).

**Table 2.** Mean predictive ability and Deviance Information Criterion of growth traits of the *S. platyclados* population, before thinning, derived using all SNPs and sets meeting GWAS-based criteria obtained with indicated Bayesian models (Bayesian LASSO, Bayesian Ridge Regression, Bayes A, Bayes B and Bayes C).

| Threshold | Model | Stem Diameter_2014 | | Total Height_2014 | |
|---|---|---|---|---|---|
| | | PredAbi | DIC | PredAbi | DIC |
| All SNPs | BL | 0.158 | 1182.032 | 0.075 | 459.510 |
| | BRR | 0.145 | 1181.166 | 0.043 | 458.551 |
| | Bayes A | 0.150 | 1181.264 | 0.068 | 458.586 |
| | Bayes B | 0.155 | 1182.357 | 0.058 | 458.835 |
| | Bayes C | 0.153 | 1180.895 | 0.050 | 459.336 |
| $-\log_{10}(P) > 0.5$ | BL | 0.080 | 1083.879 | −0.005 | 357.158 |
| | BRR | 0.075 | 1078.096 | **−0.003** | **352.719** |
| | Bayes A | 0.078 | 1080.453 | −0.005 | 355.732 |
| | Bayes B | 0.075 | 1083.867 | −0.005 | 358.664 |
| | Bayes C | 0.078 | 1079.938 | −0.005 | 355.619 |
| $-\log_{10}(P) > 0.75$ | BL | 0.088 | 1068.282 | −0.005 | 342.287 |
| | BRR | **0.085** | **1061.611** | −0.010 | 337.228 |
| | Bayes A | 0.083 | 1064.498 | −0.008 | 340.134 |
| | Bayes B | 0.088 | 1068.903 | −0.010 | 342.533 |
| | Bayes C | 0.088 | 1065.340 | −0.010 | 340.017 |
| $-\log_{10}(P) > 1$ | BL | 0.080 | 1061.138 | −0.025 | 332.349 |
| | BRR | 0.080 | 1055.108 | −0.025 | 328.483 |
| | Bayes A | 0.083 | 1057.265 | −0.025 | 329.535 |
| | Bayes B | 0.088 | 1062.515 | −0.030 | 333.926 |
| | Bayes C | 0.085 | 1060.258 | −0.028 | 332.131 |
| $-\log_{10}(P) > 1.25$ | BL | 0.065 | 1065.320 | −0.003 | 326.738 |
| | BRR | 0.068 | 1061.097 | −0.008 | 322.226 |
| | Bayes A | 0.068 | 1062.260 | −0.008 | 324.692 |
| | Bayes B | 0.075 | 1067.433 | −0.010 | 329.869 |
| | Bayes C | 0.075 | 1065.551 | −0.013 | 327.581 |
| $-\log_{10}(P) > 1.5$ | BL | 0.083 | 1069.619 | −0.065 | 335.293 |
| | BRR | 0.075 | 1066.736 | −0.065 | 331.438 |
| | Bayes A | 0.080 | 1067.921 | −0.068 | 332.917 |
| | Bayes B | 0.088 | 1072.385 | −0.068 | 338.721 |
| | Bayes C | 0.088 | 1070.623. | −0.068 | 337.967 |

The highest value in each trait is in bold. PredAbi, prediction ability; DIC, deviance information criterion; BL, Bayesian LASSO; BRR, Bayesian Ridge Regression; Bayes A, Bayesian A; Bayes B, Bayesian B; Bayes C, Bayesian C.

**Table 3.** Mean predictive ability and Deviance Information Criterion of growth, branch and wood quality traits of the *S. platyclados* population, after thinning, derived using all SNPs and sets meeting GWAS-based criteria obtained with indicated Bayesian models (Bayesian LASSO, Bayesian Ridge Regression, Bayes A, Bayes B and Bayes C).

| Threshold | Model | Stem Diameter_2017 | | Total Height_2017 | | Clear Bole Height | | Branch Angle | | Branch Diameter Ratio | | Wood Density | | Wood Stiffness | |
|---|---|---|---|---|---|---|---|---|---|---|---|---|---|---|---|
| | | PredAbi | DIC | PredAbi | DIC | PredAbi | DIC | PredAbi | DIC | PredAbi | DIC | PredAbi | DIC | PredAbi | DIC |
| All SNPs | BL | 0.260 | 1526.615 | 0.165 | 574.750 | 0.005 | 1184.544 | −0.080 | 381.601 | 0.230 | 848.161 | 0.233 | 995.024 | 0.170 | 2083.834 |
| | BRR | 0.255 | 1520.588 | 0.153 | 563.733 | −0.023 | 1185.003 | −0.078 | 385.408 | 0.220 | 839.370 | 0.223 | 992.683 | 0.165 | 2080.137 |
| | Bayes A | 0.258 | 1523.512 | 0.148 | 556.107 | −0.013 | 1184.597 | −0.088 | 382.937 | 0.225 | 842.294 | 0.230 | 995.637 | 0.165 | 2081.160 |
| | Bayes B | 0.255 | 1522.414 | 0.148 | 564.285 | −0.013 | 1184.720 | −0.080 | 383.137 | 0.220 | 843.542 | 0.228 | 993.401 | 0.173 | 2081.106 |
| | Bayes C | 0.255 | 1522.825 | 0.153 | 565.823 | −0.005 | 1185.273 | −0.078 | 383.844 | 0.230 | 840.545 | 0.220 | 993.448 | 0.165 | 2081.010 |
| $-\log_{10}(P) > 0.5$ | BL | 0.233 | 1386.984 | 0.105 | 411.442 | −0.070 | 1086.862 | −0.063 | 308.514 | 0.188 | 678.869 | 0.180 | 872.559 | 0.105 | 1978.435 |
| | BRR | 0.235 | 1380.652 | **0.103** | **404.501** | **−0.070** | **1082.063** | −0.063 | 303.339 | 0.185 | 671.133 | **0.180** | **864.492** | 0.105 | 1971.373 |
| | Bayes A | 0.238 | 1382.296 | 0.103 | 406.776 | −0.073 | 1084.354 | −0.063 | 305.621 | 0.185 | 673.787 | 0.178 | 866.017 | 0.108 | 1973.105 |
| | Bayes B | 0.235 | 1387.565 | 0.105 | 411.455 | −0.073 | 1088.692 | −0.063 | 309.565 | 0.188 | 679.985 | 0.178 | 870.848 | 0.105 | 1975.335 |
| | Bayes C | 0.240 | 1383.941 | 0.100 | 407.718 | −0.075 | 1085.632 | −0.065 | 306.744 | 0.185 | 673.841 | 0.178 | 867.477 | 0.108 | 1973.510 |
| $-\log_{10}(P) > 0.75$ | BL | 0.248 | 1360.589 | 0.068 | 385.929 | −0.068 | 1068.755 | −0.060 | 291.276 | 0.188 | 656.869 | 0.178 | 851.592 | 0.115 | 1955.416 |
| | BRR | 0.246 | 1358.405 | 0.069 | 382.413 | −0.068 | 1066.210 | −0.058 | 289.408 | 0.186 | 652.858 | 0.178 | 848.949 | 0.114 | 1954.200 |
| | Bayes A | 0.245 | 1357.710 | 0.068 | 380.692 | −0.073 | 1065.465 | −0.060 | 290.056 | 0.185 | 651.084 | 0.175 | 848.884 | 0.115 | 1953.466 |
| | Bayes B | 0.243 | 1361.771 | 0.070 | 387.482 | −0.068 | 1070.474 | −0.060 | 293.956 | 0.183 | 656.417 | 0.175 | 852.915 | 0.120 | 1955.575 |
| | Bayes C | 0.248 | 1358.770 | 0.068 | 382.314 | −0.073 | 1067.379 | −0.060 | 290.647 | 0.183 | 653.112 | 0.175 | 848.861 | 0.110 | 1954.110 |
| $-\log_{10}(P) > 1$ | BL | 0.238 | 1358.965 | 0.055 | 386.086 | −0.080 | 1056.541 | −0.078 | 276.519 | 0.205 | 654.709 | 0.163 | 839.495 | 0.120 | 1940.670 |
| | BRR | **0.248** | **1353.824** | 0.058 | 378.666 | −0.083 | 1050.718 | −0.073 | 272.066 | 0.203 | 647.174 | 0.163 | 833.481 | 0.118 | 1937.816 |
| | Bayes A | 0.248 | 1355.459 | 0.055 | 382.559 | −0.083 | 1053.686 | −0.080 | 274.796 | 0.205 | 650.642 | 0.165 | 835.555 | **0.118** | **1937.764** |
| | Bayes B | 0.240 | 1359.282 | 0.053 | 388.164 | −0.080 | 1058.229 | −0.090 | 279.362 | 0.200 | 655.712 | 0.158 | 841.589 | 0.123 | 1940.677 |
| | Bayes C | 0.250 | 1356.864 | 0.055 | 384.505 | −0.080 | 1055.776 | −0.083 | 277.321 | 0.205 | 652.291 | 0.165 | 837.947 | 0.120 | 1938.827 |
| $-\log_{10}(P) > 1.25$ | BL | 0.205 | 1362.100 | 0.073 | 387.631 | −0.083 | 1051.808 | −0.058 | 273.413 | 0.210 | 656.321 | 0.145 | 847.827 | 0.110 | 1940.075 |
| | BRR | 0.208 | 1357.921 | 0.080 | 381.487 | −0.078 | 1047.357 | **−0.053** | **268.808** | **0.210** | **650.858** | 0.150 | 842.105 | 0.108 | 1936.906 |
| | Bayes A | 0.208 | 1359.631 | 0.075 | 384.428 | −0.080 | 1049.106 | −0.058 | 271.239 | 0.208 | 652.110 | 0.143 | 844.628 | 0.110 | 1937.296 |
| | Bayes B | 0.203 | 1363.105 | 0.075 | 390.441 | −0.090 | 1055.538 | −0.070 | 276.558 | 0.210 | 656.525 | 0.140 | 851.635 | 0.115 | 1942.074 |
| | Bayes C | 0.200 | 1362.139 | 0.075 | 386.306 | −0.083 | 1053.154 | −0.065 | 274.552 | 0.210 | 654.503 | 0.140 | 847.205 | 0.118 | 1940.165 |
| $-\log_{10}(P) > 1.5$ | BL | 0.208 | 1385.003 | 0.048 | 398.017 | −0.088 | 1064.260 | −0.090 | 304.395 | 0.185 | 670.608 | 0.150 | 859.403 | 0.095 | 1944.753 |
| | BRR | 0.213 | 1378.850 | 0.045 | 392.367 | −0.088 | 1060.175 | −0.095 | 278.907 | 0.185 | 663.340 | 0.153 | 856.001 | 0.095 | 1941.782 |
| | Bayes A | 0.208 | 1381.038 | 0.045 | 394.956 | −0.090 | 1061.657 | −0.098 | 281.267 | 0.185 | 666.509 | 0.140 | 856.859 | 0.095 | 1941.671 |
| | Bayes B | 0.210 | 1388.738 | 0.045 | 401.640 | −0.095 | 1068.351 | −0.105 | 287.094 | 0.183 | 672.285 | 0.140 | 860.766 | 0.100 | 1946.401 |
| | Bayes C | 0.210 | 1385.915 | 0.048 | 399.274 | −0.100 | 1067.008 | −0.103 | 285.716 | 0.185 | 668.911 | 0.143 | 860.191 | 0.100 | 1946.039 |

The highest value in each trait is in bold. PredAbi, prediction ability; DIC, deviance information criterion; BL, Bayesian LASSO; BRR, Bayesian Ridge Regression; Bayes A, Bayesian A; Bayes B, Bayesian B; Bayes C, Bayesian C.

Our results clearly show that the prediction accuracies varied depending on the traits and GWAS-based threshold *p* value for including markers in the analysis. When the $-\log_{10}(P) > 0.5$ threshold was used for significant markers, the prediction accuracies before thinning were 0.075 to 0.080 for stem diameter and −0.005 to −0.003 for height (Table 2). The prediction accuracies after thinning with this threshold were 0.233 to 0.240 for stem diameter, 0.100 to 0.105 for height, −0.075 to −0.070 for clear bole, −0.065 to −0.063 for branch angle, 0.185 to 0.188 for branch diameter ratio, 0.178 to 0.180 for wood density and 0.105 to 0.108 for wood stiffness. Using the $-\log_{10}(P) > 0.75$ threshold resulted in prediction accuracies of 0.083 to 0.088 and -0.010 to -0.005 for stem diameter and height before thinning, respectively. After thinning, the prediction accuracies were 0.243 to 0.248 for stem diameter, 0.068 to 0.070 for height, −0.073 to −0.068 for clear bole, −0.060 to −0.058 for branch angle, 0.183 to 0.188 for branch diameter ratio, 0.175 to 0.178 for wood density and 0.110 to 0.120 for wood stiffness. Similar variations were also obtained with other threshold *p* values (Table 3).

In addition, changes in the GWAS-based threshold substantially changed the prediction accuracies, as measured by DIC values (Table 3). Prediction accuracy for stem diameter was highest (and DIC lowest: 1353.824) with the $-\log_{10}(P) > 1$ (0.248) and Bayesian Ridge Regression (BRR) model. For this trait, the prediction accuracy decreased when other $-\log_{10}(P)$ thresholds were used. Wood stiffness also showed the highest prediction accuracy with the $-\log_{10}(P) > 1$ (0.118) using the Bayes A model. In contrast, prediction accuracy for height was highest (and DIC lowest: 404.501) with the $-\log_{10}(P) > 0.5$ (0.103) and BRR model. Prediction accuracies for clear bole were close to zero with all thresholds, but the highest accuracy was found in $-\log_{10}(P) > 0.5$ (−0.070) using the BRR model (DIC: 1082.063). The prediction accuracy for wood density was also highest with the $-\log_{10}(P) > 0.5$ (0.180) and BRR model (DIC: 864.492). Furthermore, the highest prediction accuracies for branch angle and branch diameter ratio (−0.053 and 0.210, respectively) were obtained with the $-\log_{10}(P) > 1.25$ using BRR model, which thus mostly provided better prediction accuracies than the other Bayesian models.

## 3.4. Prediction Accuracies Based on the Machine Learning Methods

Using all SNPs, the BART method provided higher genomic prediction accuracies for stem diameter both before and after thinning (0.169 and 0.270, respectively) than XgBoost (0.132 and 0.198, respectively) and RF (0.161 and 0.238, respectively). BART also provided higher prediction accuracy for clear bole height, branch diameter ratio, wood density and wood stiffness (0.030, 0.212, 0.226 and 0.155, respectively) than the other two models. RF provided higher accuracy (−0.113) than XgBoost (−0.116 and BART (−0.126) for branch angle. XgBoost provided the highest prediction accuracy for height after thinning (0.169) (Table 4 and Table S4).

**Table 4.** Mean predictive ability of growth, branch and wood quality traits of the *S. platyclados* population derived using all SNPs and the best sets meeting GWAS-based criteria obtained with indicated Machine Learning models (RF, XgBoost and BART).

| Threshold | Model | Stem Diameter_2014 | Total Height_2014 | Stem Diameter_2017 | Total Height_2017 | Clear Bole Height | Branch Angle | Branch Diameter Ratio | Wood Density | Wood Stiffness |
|---|---|---|---|---|---|---|---|---|---|---|
| All SNPs | RF | 0.161 | 0.031 | 0.238 | 0.164 | −0.021 | −0.113 | 0.195 | 0.212 | 0.138 |
| | XgBoost | 0.132 | −0.016 | 0.198 | 0.169 | −0.069 | −0.116 | 0.182 | 0.147 | 0.155 |
| | BART | 0.169 | 0.084 | 0.270 | 0.138 | 0.030 | −0.126 | 0.212 | 0.226 | 0.155 |
| GWAS−based threshold | RF | **0.166** | 0.017 | **0.244** | **0.173** | −0.022 | −0.090 | **0.204** | **0.205** | **0.130** |
| | XgBoost | 0.105 | 0.001 | 0.172 | 0.125 | **0.035** | −0.131 | 0.156 | 0.112 | 0.075 |
| | BART | 0.151 | **0.079** | 0.203 | 0.108 | −0.016 | **−0.065** | 0.187 | 0.158 | 0.122 |

The highest value in each trait is in bold. RF, Random Forest; XgBoost, Extreme Gradient Boosting; BART, Bayesian Additive Regression Tree.

Using subsets of SNPs meeting GWAS-based significance criteria resulted in lower prediction ability for diameter than using all markers (0.244, 0.172 and 0.203 estimated by RF, XgBoost and BART models, respectively). This trend was also found in height traits. The prediction accuracies for clear bole were close to zero (−0.022 to 0.035). For the branch angle, BART (−0.065) provided the highest prediction accuracy, and XgBoost the lowest (−0.131). For the diameter ratio, RF and BART provided higher accuracies (0.204, 0.187) than XgBoost (0.156). Moreover, using markers meeting GWAS-based criteria, RF provided the highest accuracy for wood density (0.205) and wood stiffness (0.130), respectively (Table 4).

### 3.5. Genomic Heritability

As shown in Table 5, the estimated genomic heritability varied, depending on the traits. In addition, genomic heritability estimated using all markers were much lower than those estimated using sets of markers meeting GWAS-based significance criteria. Generally, using selected markers, genomic heritability was higher for growth and wood quality traits (0.573, 0.615, and 0.516 for stem diameter, height and wood density, respectively) than for branch quality traits (0.342 for branch angle). However, the genomic heritability of the branch diameter ratio (0.526) was similar to the estimates for wood density and stem diameter.

**Table 5.** Genomic heritability of growth, branch and wood quality traits of *S. platyclados* half-sib progeny population obtained using all SNPs and selected markers based on the highest $-\log_{10}(P)$.

| Traits | Genomic Heritability | |
| --- | --- | --- |
| | **All SNPs** | **Selected Marker** |
| Stem diameter_2014 | 0.247 | 0.475 |
| Total height_2014 | 0.224 | 0.457 |
| Stem diameter_2017 | 0.313 | 0.573 |
| Total height_2017 | 0.292 | 0.615 |
| Clear bole height | 0.211 | 0.466 |
| Branch angle | 0.190 | 0.342 |
| Branch diameter ratio | 0.337 | 0.526 |
| Wood density | 0.263 | 0.516 |
| Wood stiffness | 0.252 | 0.473 |

## 4. Discussion

### 4.1. Population Structure

Population structure is one of various factors that may strongly influence LD, leading to confounding effects and false positives or spurious associations between traits and marker alleles. Thus, population structure should be properly accounted for in marker-trait association analysis [62,63]. We found no distinct spatial clusters in PCA analysis, indicating that the *S. platyclados* alleles were distributed without strong structure, which is highly beneficial for GWAS resolution and genomic prediction accuracy [60,64].

The weakness of population structure is probably due to the progenies originating from sources with similar genetic backgrounds, since the seeds were collected solely from mother trees in the PT-SBK (which covers 147,600 ha in Central Kalimantan). Relatively low levels of genetic differentiation have also been previously detected within populations of two other *Shorea* species in Borneo, *S. parvifolia* (FST < 0.169) [65] and *S. curtisii* (FST < 0.035) [66]. In addition, other studies have clearly differentiated two groups of *S. platyclados* populations, called the "western Malaysian" (Sumatran and Peninsular Malaysian) and "eastern Malaysian" (Bornean) groups due to a long period of isolation and geographic separation by the South China Sea [67,68]. Similar divergence has also been detected in *S. leprosula* populations by expressed sequence tag-based simple sequence repeat (EST-SSR) and chloroplast DNA polymorphism (cpDNA) sequencing analyses [69].

### 4.2. Detection of Significant Markers by GWAS

A major concern in GWAS is the detection of falsely positive significant SNPs that arising from population structure and family relatedness, which can result in assignation of strong association between variants and traits that are not causally linked [63]. Although our present study showed a low deviation between observed and expected *P* values in the quantile-quantile plot, however, it seems that the Q+K model did not provide better control for false positives than that the Q, K or even the naïve model. This may the effect of a very weak structure in our population and the individuals that come from the half sib-family are also closely related. Therefore, including population structure and kinship in the MLM model did not result a significantly differences from the GLM model. In general, the MLM model in the analysis may increase power to detect true associations than ignoring these two factors which may result spurious associations. In our study, *S. platyclados* population has kinship structure that comes from half-sib, thus, we applied the Q+K model in the GWAS analysis.

GWAS includes application of a stringent threshold for inclusion of SNPs based on multiple testing false discovery rate (FDR) [70–72]. We found that GWAS had low power for detecting significant SNPs, since very few linked to the focal traits were detected when high-stringency FDR criteria were applied. This is consistent with findings of previous GWAS of tree species [22,73,74]. For example, [22,74] identified just one SNP associated with volume growth in *E. pellita* and no significant associations in *E. benthamii*. Thus, GWAS involving application of such multiple testing FDR criteria might not be appropriate for detecting small effects of multiple SNPs [75].

Although false discovery tests are important for controlling type I errors (false positives), if there are numerous SNPs in strong LD, they could cause substantial loss of statistical power and increase risks of missing true associations, or almost certainly inflate type II errors (false negatives) [76]. Surprisingly, our study revealed high and slowly decaying LD, extending more than 145 kb, in comparison to the reported decay within just 500 bp in European aspen (*Populus tremula*) [77] and 2 kb in loblolly pine (*Pinus taeda*) [78]. However, too few markers may have been used in the cited studies (384 and 288 SNPs, respectively) to obtain accurate LD estimates. Recent studies have detected more extensive LD, e.g., up to 10–12 kb in *E. globulus* [79], 16–34 kb in poplar (*P. trichocarpa*) [80] and 65–110 kb in *C. japonica* [81]. A high proportion of SNPs in non-coding regions of the genome may also mask the true extent of LD, as they provide lower estimates of recombination rates between loci than SNPs in coding regions [81]. In addition, the FDR threshold for significant markers should be less stringent for species with high LD than for species with low LD [76]. Therefore, it is important to select an appropriate threshold for significant markers to differentiate true positives from false positives and false negatives in GWAS. Under the high LD, although 5900 SNPs still has possibility not to be able to detect any of QTLs due to the sparse distribution, we, thus, applied five $-\log_{10}(P)$ thresholds from stringent to weak to selected markers that were significantly associated with the focal traits for GS. This empirical approach has also been applied in previous studies of *C. japonica* and *Eucalyptus* [33,82], and it identified some SNPs that appeared to be significantly related to phenotypic traits. Under the high LD of the population, we expected that this approach may identify optimal balance to select makers. If we use a very stringent threshold, we may lose the chance to detect significant association with QTLs. However, high LD may help to detect association (not significant in FDR and Bonferroni correction) with QTLs by increasing false positive.

Another factor that may have contributed to the low detection power of GWAS is that numerous loci may weakly influence quantitative traits of *S. platyclados* (such as those examined here), as in other forest tree species where the most commercially important traits are usually under polygenic quantitative control [55,78,83]. High frequencies of such minor QTLs will inevitably impair the power of GWAS to detect significant SNPs [12,22,75].

### 4.3. Genomic Predictions Based on All SNPs and GWAS-Based Thresholds

GS is not affected by the limitation of GWAS linked to the problem of detecting markers that are significantly associated with polygenic traits, because it exploits the predictive power of large numbers

of markers simultaneously across the whole genome [19]. Our study revealed that it provided similar genomic prediction abilities using all SNPs and sets meeting various GWAS-based criteria. However, the variance of Genomic Estimated Breeding Values (GEBVs) resulting from GS based on all SNPs was lower than the actual phenotype variance. Thus, use of all markers may result in sub-optimal models for predicting phenotypes. Similarly, using SNPs selected according to GWAS-based criteria provided much better genomic heritability estimates than using all SNPs. Together with findings by [25] that showed that using a panel of 38,082 SNPs and subsets of 3000 selected SNP markers provided similar genomic prediction accuracy for traits of cattle, these results suggest that increasing numbers of SNPs in a genomic prediction model does not always increase accuracy. This may be at least partly because the additional background noise from non-related markers impairs the prediction quality, by introducing more prediction errors than a smaller number of markers that capture the main effects of truly relevant SNPs, SNP-SNP interactions and non-linear relationships [25].

### 4.4. Genomic Prediction Accuracies of Bayesian and Machine Learning Methods

Our estimated prediction accuracies for growth traits of *S. platyclados* were similar to those obtained for *E. benthamii* (DBH 0.153–0.162, height 0.019–0.025) and *C. japonica* (DBH 0.033–0.209, height 0.026–0.114) populations in the North and South Kanto region of Japan. However, it was slightly lower than those reported on *E. pellita* (DBH 0.427–0.438, height 0.339–0.341), *C. japonica* (DBH 0.505, height 0.432) populations in the Kyushu region of Japan, *Pinus taeda* (DBH 0.46, for height 0.38) and *E. globulus* (DBH 0.17–0.45, height 0.21–0.44) [22,23,33,79]. We obtained similar prediction accuracies for wood density to reported values for loblolly pine (0.112–0.226 and 0.20–0.23, respectively), but much lower accuracy for wood stiffness (0.075–0.155 and 0.39–0.42, respectively) [23].

Moreover, our study showed that the five models provided very similar estimates and prediction accuracies, in accordance with the previous studies using material from *E. pellita*, *E. benthamii*, *Pinus taeda* and *Picea mariana* [22,84]. The BRR was the best model, in terms of DIC, which are correlated with residual variance, and thus, indicate models' fitting to the data used to construct them [85,86]. The statistical models are based on different assumptions about markers' numbers and effects, which may affect GEBV values and the accuracy of predictions obtained from different data sets. The assumptions underlying BRR (that there are many QTLs with small effects) seem appropriate for our material. In contrast, Bayesian LASSO and Bayes A models are based on assumptions that many QTLs have small effects and a few QTLs have large effects, and Bayes B models on assumptions that many QTLs have no effects and few have some, possibly large, effects [19,87].

Furthermore, prediction accuracies obtained from all Bayesian and Machine Learning methods were trait-dependent, as found by [60,88]. However, all the Machine Learning methods provided better prediction accuracies for all focal traits of *S. platyclados* than the Bayesian methods. This suggest that Machine Learning is an appropriate approach for *S. platyclados*. A previous study found that RF was an efficient method for identifying a subset of SNPs linked to candidate genes affecting growth traits of beef cattle [25]. Similarly, we found that of the tested Machine Learning methods, RF provided the highest prediction accuracies for diameter, height, branch diameter ratio, wood density and wood stiffness (and thus both growth, branch and wood quality traits). This may be due to its ability to capture effects of all markers, including those with weak effects, enabling complex correlations and interactions among markers to contribute to the model fit, and thus, provide accurate, simple and complex regressions [89]. Moreover, BART models tended to provide the best predictions in wood quality traits obtained using all SNPs, possibly because of their ability to handle all types of genetic effects of SNPs. Similarly, [25] found that BART models provided better genomic predictions for pig traits than other methods, i.e. Random Forest (RF), Bayesian Least Absolute Shrinkage and Selection Operator (BLASSO), Genomic Best Linear Unbiased Prediction (GBLUP) and Reproducing Kernel Hilbert Space (RKHS). XgBoost provided the best performance for predicting the clear bole height in our study, but all the models' prediction accuracies for this trait were poor.

*4.5. Genomic Heritability*

Since estimating the heritability of growth and wood traits in forest tree species using pedigree information is time-consuming, expensive and not very accurate, genomic heritability estimation is a better option. To our knowledge, this is the first report on the estimation of heritability in Dipterocarps using genome-wide markers. Surprisingly, the estimated heritability for growth and wood quality traits of *S. platyclados* were much higher than those reported for height and wood density of Norway spruce (0.15 and 0.34, respectively) [21] and height of *E. benthamii* and *E. pellita* (0.190 and 0.455, respectively) [22]. The differences in heritability values obtained for the same trait among these studies might be due to differences in the estimation methods, numbers of families and environmental interactions. Like the two cited studies, we also found that heritability of wood density was as higher as found in diameter and height growth, in accordance with general tendencies for wood properties of tropical hardwoods to have higher heritability than their growth traits [90]. Moreover, gene-environment ($G \times E$) interaction effects are generally strong for traits with low heritability and weak for traits with high heritability. Similarly, there are indications that $G \times E$ effects are extremely weak for wood quality traits in pine [91,92] and spruce [93].

*4.6. Factors Affecting the Accuracy of GS of S. platyclados*

The success of genomic selection primarily depends on predictive ability, which is influenced by several factors, such as the heritability of focal traits, training population size, marker density, the relatedness of training and validation populations, extent of LD and the statistical methods used [15,94]. Our study indicates that heritability has a minor impact on the accuracy of GS. Similarly, a previous study found that GS accuracy increased, by 10–20%, with the increase in heritability from 0.2 to 0.6 [15]. Therefore, a larger training population is needed to achieve the same level of predictive ability for traits with high heritability than for traits with low to moderate heritability, such as those considered in this study [95]. We had less than 300 individuals in the training sets, which probably impaired the prediction accuracy, as selection accuracy can reportedly rapidly rise with increases up to 1000 individuals [15]. For example, a study on *Picea mariana* found that training populations of 330 to 740 trees resulted in good prediction accuracies, but reducing the training set from 740 to 184 reduced GS models' prediction accuracy by 50% [85]. Increasing training sets sizes is expected to increase prediction accuracy by reducing bias and reducing the variance of marker effect estimates [96,97].

Increasing the marker density also has an impact because it increases chances of QTLs being in LD with a marker, which increases the prediction accuracy [98]. Thus, the relatively low marker densities used in this study may have been insufficient to capture QTLs and markers in LD, although LD decayed slowly. Moreover, the half-sib families used in our study may have contributed to the relatively low prediction ability. Generally, the prediction ability is highest for full-sib families and lowest for populations with no family structure, indicating (unsurprisingly) that GS modeling is most efficient for closely related individuals. This trend has been confirmed, *inter alia*, by studies of patterns in black spruce (*Picea mariana*) [85], white spruce [29] and beef cattle [25].

*4.7. Potential Utility of GWAS and GS for Breeding Tropical Tree Species*

Relative to the long history of breeding improvement in conifer species, such as *Pinus taeda* (fourth generation) [99], *P. radiata* (third generation) [100] and *C. japonica* (second generation) [33], breeding Dipterocarpaceae as one of the tropical rainforest tree species is a relatively new endeavor. In this case, tree breeders face the difficulty to select mother trees from natural forest (characterized by the difference of sizes and ages) due to less proper guideline for the selection of mother trees. Initially, we applied visual evaluation to selected trees that showed phenotypically superior in the natural forest. Furthermore, tree breeders tended to select mother trees in terms of a few phenotypic traits, such as stem diameter and tree height, as easily measurable characteristics [101]. At that time, we considered these two traits to be important to improve the growth traits that are associated with the yield of timber

product. After screening the selected mother trees, the breeders now needed to establish a progeny trial to identify the superior parental based on the performance of their offspring.

A progeny trial in a tree breeding program usually covers a large area. An important factor in the selection and assessment of phenotypic traits from large field areas was microenvironmental heterogeneity, such as soil fertility, solar radiation, water availability, etc. Thus, while many traits were assessed in the different environment, in practice, tree breeders sometimes have difficulty in evaluating the phenotype. Although more replications of the plot are established, selection was often biased. Beside the practical difficulties in phenotypic assessment, phenotypic evaluation in the tropical tree species is by far the most expensive activity in tree breeding programs, not only in terms of data collection, but also in terms of trial establishment and maintenance, analyzing data and maintaining records [102].

The results suggest that GWAS will probably be inefficient for detecting significant SNPs related to the focal traits, but the genome-wide DNA markers developed for GWAS can be used directly for GS modeling [32,103,104]. Based on the initial prediction accuracy, information obtained from the F1 generation of the *S. platyclados* population could be applied in GS for the next generation using seedling material. This implies that we only need sequencing data showing young seedlings' genotypes, without data on phenotypic characteristics [105]. Such early selection would enable breeders to accelerate tree improvement by reducing the need to phenotype large numbers of adult trees in field trials [8]. In addition, the high genome-wide throughput, low ascertainment bias and low genotyping cost of next-generation sequencing [12] will support the routine use of GS in tropical forest tree breeding programs.

## 5. Conclusions

This is the first empirical study of both GWAS and GS in important tropical timber species in rainforests of Southeast Asia. Using a full set of genome-wide markers, our study revealed fast and slow decaying LD, extending more than 145 kb. GWAS will probably be inefficient for detecting significant SNPs related to the focal traits, but GWAS assisted GS seems to be a promising approach for *S. platyclados* breeding programs, especially in conjunction with Machine Learning methods. Moreover, the generally high genomic heritability estimates for growth and wood quality was a precious finding for the very early breeding history of tropical rainforest tree species, since unavailability materials for estimate the pedigree heritability directly. It is also indicated that selective breeding for these traits individually could be very effective, especially for increasing the diameter growth, branch diameter ratio and wood density simultaneously.

**Supplementary Materials:** The following are available online at http://www.mdpi.com/1999-4907/11/2/239/s1, Figure S1: Manhattan and Q-Q plots for genome-wide association of *S. platyclados* growth traits (before thinning) with SNPs markers of a. naïve mode; b. Q model; c. K model. The quantile-quantile plot indicates the fitness between expected (red lines) and observed *p*-values (black dots). Figure S2. Manhattan and Q-Q plots with no structure or kinship effect (naïve model); Figure S3: Manhattan and Q-Q plots with the effects of population structure (Q model); Figure S4: Manhattan and Q-Q plots with the effects of kinship relationship (K model); Table S1: Genetic diversity statistics of *S. platyclados* half-sib progeny population. *Na*, number of alleles; *Ne*, number of effective alleles; *I*, Shannon's Index; *h*, diversity index; *uh*, unbiased diversity index; *PPL*, percentage of polymorphic loci; Table S2: $-\log_{10}(P)$ and Q (FDR) values for GWAS analysis of *S. platyclados* half-sib progeny population; Table S3: Predictive ability and DIC of *S. platyclados* traits obtained with indicated Bayesian models. The analysis performed in four-fold cross-validation using all SNP markers and selected SNP markers with high $-\log_{10}(P)$ value threshold. PredAbi, prediction ability; DIC, deviance information criterion; BL, Bayesian LASSO; BRR, Bayesian Ridge Regression; Bayes A, Bayesian A; Bayes B, Bayesian B; Bayes C, Bayesian C; Table S4: Predictive ability for *S. platyclados* traits obtained with indicated Machine Learning methods. The analysis performed in four-fold cross-validation using all SNPs and selected $-\log_{10}(P)$ threshold. RF, Random Forest; XgBoost, Extreme Gradient Boosting; BART, Bayesian Additive Regression Tree.

**Author Contributions:** Conceptualization, N.T. and Y.T.; methodology, N.T. and K.U.; software, S., N.T. and K.U.; validation, S., N.T., K.U. and Y.T.; formal analysis, S., N.T. and K.U.; investigation, S., W., S.I., M.N. and N.T.; resources, M.N., W., S.I., R.S., K.K.S.N. and S.L.L.; data curation, N.T.; writing—original draft preparation, S.; writing—review and editing, S., N.T., Y.T. and K.U.; visualization, S.; supervision, N.T. and Y.T.; project

administration, N.T.; funding acquisition, N.T., Y.T., M.N., W. and S.I. All authors have read and agreed to the published version of the manuscript.

**Funding:** This study was part of the doctoral dissertation of Sawitri, a recipient of the MEXT Japan scholarship (grant number 173162). This research was funded by a JIRCAS-UGM joint project entitled "Enhancement of productivity using genetic resources in tropical rainforest and development of carbohydrate usage from unutilized biomass in Indonesia" (grant number a1C402b; JIRCAS) and the APC was funded by the JIRCAS-UGM joint project.

**Acknowledgments:** We would like to thank the technical staff of the progeny trial in PT-Sari Bumi Kusuma for their invaluable contributions during the sample collection and K. Yamashita, S. Ohashi and H. Aiso for their instruction to measure wood density and stiffness. We would like to also thank H. Iwata for his advice to improve GWAS and GS analyses and H. Mori for his supports to conduct spatial analysis.

**Conflicts of Interest:** The authors declare no conflict of interest.

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
