# Peer review of "Potential of Genome-Wide Association Studies and Genomic Selection to Improve Productivity and Quality of Commercial Timber Species in Tropical Rainforest, a Case Study of Shorea platyclados"

_forests, doi:10.3390/f11020239_

Round 1

Reviewer 1 Report

The authors have addressed all the issues I raised in my previous review. I have no further questions at this time. 

Reviewer 2 Report

Potential of Genome-Wide Association Studies and Genomic Selection to improve productivity and quality of commercial timber species in tropical rainforest, a case study of Shorea platyclados

I would like to appreciate the efforts of the Authors to improve and re-submit the manuscript.  The paper is much better access for readers to understand the methods, elaboration of results and conclusions are derived in logical consistency to the scientific background of introduction and proposed manner of utilised methods.

This manuscript is a resubmission of an earlier submission. The following is a list of the peer review reports and author responses from that submission.

Round 1

Reviewer 1 Report

This manuscript is written very well. However, I have some comments

PPCA showed wide distribution of accessions. I consider this as stratified population. In addition, if you present population structure analyzed using Bayesian methods to confirm that the population is not admixed to support your results.   Though machine learning tools are applied, the SNP number for each trait presented in Table 1 at the negative log10 1.5 is showing higher number of SNPs. There are only about 5K SNPs in total. I wish to see top ten associations. Can you increase this stringency or stick with the SNPs with Bonferoni correction. Can you also present any candidate genes harboring these SNPs? SNPs may be listed in Table 5 for which heritability is estimated Manhatten plots show SNP effects are quite varied. Is this because of lesser population size?

Reviewer 2 Report

Review of Sawitri et al “Potential of genome.wide associaton studies and …..”

Sawitri et al use SNP data obtained from ddRADseq to perform both genome-wide association studies (GWAS) and genomic selection (GS) in the tropical forest tree Shorea platyclados. The GWAS studies failed to detect any SNPs that were significantly associated with any of the phenotypic traits studied. The GS analyses showed that machine learning methods generally performed better than traditional, Bayesian methods, but the results were highly trait-dependent.

Lines 140-143: The “Material and Methods” section stated that 420 samples were phenotyped. How were these 420 individuals distributed among the 81 possible half.sib families? What were the mean, minimum and maximum number of offspring per family?  I’m missing some basic information on the genome of Shorea platyclados. How large is the genome, how many chromosomes do the species have?  Lines 174: Some more information on the reference genome would be useful. How contiguous in the reference genome? How many scaffolds, the average size of scaffolds? This is important background information that is needed to interpret, among other things, the LD analyses presented later in the ms.  Lines 178: Did you only obtain 337,936 sequencing reads in total across all 367 individuals? What fraction of the S. platyclados genome does this correspond to? What was the average coverage per individual? All this is important background information that is needed to interpret the results of the genotyping.  Lines 199-201: For the PCA the authors state that they thing the SNP data by selecting SNPs every 1000bp. However, the LD analyses show that LD extends over distances up to 111kb, suggesting that dropping sites closer than 1kb may not necessarily remove non-independence among sites. A more appropriate method to reduce dependence among sites would be to perform proper LD pruning, for instance as implemented in PLINK. One of the filtering steps included in the analyses of the SNP data was filtering on significant Hardy-Weinberg deviations. One of the most common reasons for deviations from HW is population structure, so filtering on HW runs the risk of reducing any evidence for population structure. How many SNPs were filtered out due to HW deviations and does including these SNPs in the PCA analyses change the results for the population structure analyses in any significant degree? Lines 220-225: From the description in the text, it appears that SNPs with up to 40% missing data were included in the final data set. Did you impute all SNPs with missing data, and if so, did you perform any validation on how accurate the imputation was across different fractions of missing data? In my experience, imputation works well for missing data up to 10-15% but the accuracy drops for higher fractions of missing data. Figure 3: This figure is really hard to read. The non-linear regression line looks more like a blur than a line to me. To me, based on the data points presented in the figure, LD appears to extend even further than indicated by the non-linear regression. Would it be possible to implement another method, such as the one advocated by Remington et al (2001, PNAS 98: 11479–11484)? The methods used by Remington et al. (which is also a non-linear regression method) has the advantage of being based on an underlying model of genetic drift and recombination, making the parameters of the model easier to interpret from an evolutionary perspective. Lines 298-302: After comparing the QQ-plots in Figure 4 and 5 with Supplementary Figures S1-S3, I’m not convinced that the Q+K model provides better control for false positives that the K or even the ‘naive’ model. I can’t see any systematic deviation from the 1:1 line in any of the supplementary plots, and using the Q+K model might in fact then be overcorrecting for the effects of population structure and kinship.  Table 5: I’m somewhat surprised that the genomic heritabilities are relatively high (and even higher using selected markers) given that the prediction accuracies, in general, were rather low and that the number of markers used was rather limited.  Is it possible to also calculate ‘regular’ heritabilities, based only on the phenotypic data and the half-sib design implemented in the field trials? If so, how do these compare with the genomic heritabilities? Also in Table 5, what criterion was used to select markers in the “Selected marker” category? This is not indicated in the legend to Table 5.  Lines 458 an onwards: An additional factor explaining the lack of significant SNPs in the GWAS analyses is also that so few SNPs were used in the analyses in the first place (~5k SNPs). Given the sparse distribution of SNPs across the genome, the likelihood of picking up strong associations seems inherently low to me.

Reviewer 3 Report

The manuscript titled: Potential of Genome-Wide Association Studies and Genomic Selection to improve productivity and quality of commercial timber species in tropical rainforest, a case study of Shorea platyclados, it touches very important problems all over the world to avoid time‐consuming, laborious methods of prediction good genotypes for future selection. The Authors tried to make more effective current breeding techniques using molecular studies measures of a full set of genome‐wide markers in the selection of economically important traits for the tree species with unknown practically genetic structure.

I think it could be good to add some more information regarding the basic genetic structure of analysed populations not only one small table in supplementary materials when the total genome structure is discussed. 

The data for single analysed populations would be more appropriate to understand the final week differentiation of populations when it is discussed. Even some comparison to other Shorea sp. would be important.

It is a little bit blurred the significance of the genetic effects of new methods achieves by too long a description of the economic importance of the species in the introduction.  This problem is not elaborated in the other part of the manuscript and it not included in the title as well. So, this part could shorter to make the paper more consistent regarding the relevance of GWAS and GS methods meaning.

The methods are presented in a very logical way, typical to scientific elaboration. The results describe the first empirical study of both GWAS and GS in Shorea platyclados, which posses original and novel character.

The quality of the tables and figures presentations is fine.

The discussion and conclusion of results were presented very clearly and properly documented by literature data.